# Influence of Intrinsic Physicochemical Properties of Agroforestry Waste on Its Pyrolysis Characteristics and Behavior

**DOI:** 10.3390/ma16010222

**Published:** 2022-12-26

**Authors:** Hui Liu, Baowei Zhao, Xin Zhang, Yin Zhang

**Affiliations:** School of Environmental and Municipal Engineering, Lanzhou Jiaotong University, Lanzhou 730000, China

**Keywords:** agroforestry wastes, intrinsic physicochemical properties, thermogravimetric analysis, pyrolysis characteristics, activation energy, reaction mechanism

## Abstract

To obtain a comprehensive understanding of the qualitative and quantitative effects of the intrinsic properties of biomass on its pyrolysis characteristics and assess the behavior of agroforestry waste, thermogravimetric analyses of three representative agroforestry wastes, namely rape (*Brassica campestris* L.) straw (RS), apple (*Malus domestica*) tree branches (ATB), and pine (*Pinus* sp.) sawdust (PS), were carried out by pyrolysis under dynamic conditions (30 to 900 °C) at different heating rates of 5, 10, and 15 °C·min^−1^. Correlation analysis showed that intrinsic physicochemical properties play distinct roles in different stages of pyrolysis. The ash content was negatively correlated with the temperature range (R_2_) of the second stage (190–380 °C) of pyrolysis. The lignin content and the amount of pyrolysis residues (RSS) were positively correlated. Kinetic triplets, including the activation energy (*Ea*), pre-exponential factor (*A*), and reaction model [*f*(*α*)], were obtained using different methods, including the Flynn–Wall–Ozawa (FWO), Freidman, Kissinger–Akahira–Sunose (KAS), and Starink methods. The mean activation energy (*Ea*[mean]) for RS, ATB, and PS calculated by the different methods ranged from 167.15 to 195.58 kJ·mol^−1^, 195.37 to 234.95 kJ·mol^−1^, and 191.27–236.45 kJ·mol^−1^, respectively. Correlation analysis of the intrinsic physicochemical characteristics and kinetic factors of agroforestry waste showed that the minimum *Ea* (*Ea*[min]) was significantly positively correlated with heat capacity (*C*_0_) and negatively correlated with thermal diffusivity (*D*). The *Ea*[mean] and the maximum value of *Ea* (*Ea*[max]) significantly positively correlated with the sum content of cellulose and lignin, indicating that the contents of cellulose and lignin determines the energy required for the pyrolysis process of agroforestry waste. The mechanism of degradation involves the diffusion model (D1, D2, and D3), the growth model (A4), and the geometrical contraction model (R3). These results indicate that the pyrolysis of agroforestry waste is a complex process due to the heterogeneity of its intrinsic physicochemical properties.

## 1. Introduction

In China, which is one of the largest agricultural counties in the world, increasing amounts of agroforestry wastes have been generated with the development of agricultural modernization and the rural economy. It has been estimated that approximately 2.1 billion tons of agricultural residues and forest residues are produced in China each year [1]. If they cannot be used and treated in a timely manner, they not only cause serious environmental pollution by releasing toxic and harmful gases [2], such as CO_X_, NO_X_, CH_4_, and SO_2_, but also result in economic losses [3]. Therefore, many promising technologies have been used to recycle and harmlessly treat agroforestry wastes. Thermochemical conversion methods, such as pyrolysis, gasification, and combustion [4], are effectively and widely used for these purposes. Pyrolysis has attracted much attention owing to its advantages, such as low reaction temperatures, fewer emissions, low cost, and simplicity of operation [5]. In addition, biomass can be converted into bio-oil [6], biochar [7], and gases through the pyrolysis process [8]. Bio-oil can be directly used as oil fuel owing to its higher volumetric energy density [9]. Biochar, as a potential material for carbon sequestration, is being increasingly utilized, and its calorific value is comparable with that of solid fuel [10,11]. The produced gases can be utilized as syngas for internal-combustion engines, in power stations, or to supply heat. Some studies have pointed out that the complete pyrolysis of agricultural waste under optimized conditions can produce approximately 47% bio-oil, 30% carbon, and 23% non-condensable gas [12], which demonstrates that the conversion of pyrolysis would be an economical and environmentally friendly way to dispose of agroforestry waste.

Thermogravimetric analysis (TGA) is widely employed to analyze the pyrolysis behaviors of biomass samples. According to the experimental data from TGA, one can obtain insights into the pyrolysis characteristics of biomass and obtain their kinetic parameters (activation energy, pre-exponential factors, and reaction mechanisms). Knowledge of the characteristics and kinetics of the biomass being subjected to pyrolysis is the basis for understanding its thermal chemical conversion mechanism, reaction rate, complexity, and the degree of difficulty achieving its reaction in order to achieve good pyrolysis control [13,14]. 

Pyrolysis of biomass can be classified as a heterogeneous chemical reaction [15]. Seemingly slight differences in heterogeneous process variables, along with heat and transport limitations, can have significant impacts on the pyrolysis behaviors and kinetic parameters, such as extra-particle factors (heating rate, particle size, pyrolysis temperature, residence time, and intra-particles factors (intrinsic physicochemical properties)). The influence of external factors on pyrolysis characteristics has been extensively studied, such as heating rate, pressure, particle size, and other factors. The maximum rate of decomposition and the whole conversion rate of pyrolysis also increases with the increasing heating rates owing to the increase in thermal energy [16]. Increasing the pressure of the batch autoclave reactor not only increases the aromaticity and crystallinity of the pyrolysis products, but also leads to an increase in the Ea of the pyrolysis process [17]. The higher particle size ranges of biomass also resulted in relatively higher average Ea arising from the heat transfer limitation between particles [18]. Thus, these studies provide an important theoretical scientific basis for tailoring extra-particle factors to achieve high production efficiency and desired products.

The intrinsic physical and chemical properties of the feedstocks also have a vital influence on the pyrolysis behavior and kinetic parameters. Among the relevant characteristics of decomposition and the kinetics of biomass, the following intrinsic properties are critical: proximate composition (water, volatile matter, ash, and fixed carbon) [19], chemical composition (cellulose, hemicellulose, and lignin) [20], ultimate analysis (C, H, N, and O) [21], and thermal properties (thermal conductivity, heat capacity, and thermal diffusivity) [22,23]. For example, the differences in proximate composition and thermophysical properties directly affect the activation energy, products, and reactor throughput of biomass pyrolysis [19]. The unique pyrolysis curve of biomass depends on its lignocellulose composition [20]. In addition, the elemental composition (C, H, O, N, and S) of biomass can be used to predict the thermodynamic properties of biomass reasonably well, such as the Gibbs free energy formation [21]. However, only a few attempts have been made to correlate the pyrolytic characteristics and kinetics of biomass with its intrinsic physical and chemical properties, and the information available on the influence of physicochemical properties of raw materials on pyrolysis behavior is therefore limited, especially regarding the pyrolytic behavior of agroforestry waste [24,25,26]. It is therefore of great significance to deeply understand the influence of the intrinsic properties of raw materials on the pyrolysis process and reaction mechanism in order to optimize the design and amplification of the pyrolysis system [27]. Furthermore, a comprehensive understanding of the qualitative and quantitative effects of these intrinsic properties on the pyrolysis characteristics and behavior of biomass can help one to directly estimate the pyrolysis rate and control the efficiency of the pyrolysis of agroforestry waste according to the intrinsic properties, rather than tedious experimental work.

In this work, the pyrolysis characteristics and kinetics of rape (*Brassica campestris* L.) straw (RS), apple (*Malus domestica*) tree branches (ATB), and pine (*Pinus* sp.) sawdust (PS) were studied as representatives of agroforestry wastes. Rapeseed, which is the primary oil crop in China, is grown in many areas of this country. Approximately 1.96–107 tons of RS is generated each year as a byproduct during the harvest season in China [28]. Apple is the primary fruit produced in northwest China. Many waste branches are generated during the harvesting and pruning season, and they need to be properly disposed of and managed. Pinewood is a vital raw material for the furniture and paper industry in China, and its processing generates sawdust. These agroforestry wastes have found no other useful applications except as firewood. Pyrolysis can be used as a promising thermochemical conversion process to effectively treat these biomasses.

Therefore, the primary objective of this study is to comprehensively analyze the intrinsic physicochemical properties of agroforestry waste, and to deeply understand the influence of intrinsic physicochemical properties of agroforestry on pyrolysis behavior and mechanism, which aims to provide a scientific basis for the efficient pyrolysis of agroforestry waste. The detailed intrinsic physicochemical characterization, including proximate analysis, ultimate analysis, lignocellulosic content, and thermophysical properties of RS, ATB, and RS were analyzed. The non-isothermal pyrolysis of RS, ATB, and PS was conducted under different heating rates in a thermogravimetric analyzer. Through a correlation analysis of the intrinsic physicochemical properties and pyrolysis characteristics of raw materials, the influence of the intrinsic physicochemical properties of biomass on the pyrolysis behavior can be explored. To obtain accurate kinetic parameters, four model-free methods were used: the Flynn–Wall–Ozawa (FWO) method, the Freidman method, the Kissinger–Akahira–Sunose (KAS) method, and the Starink method. Simultaneously, the main solid-state reaction kinetic models were carefully compared with the experimental data using the master-plot method to determine the pyrolysis reaction model of raw materials.

## 2. Materials and Methods

### 2.1. Feedstock Preparation and Characterization

RS and ATB were procured from the Longnan districts of Gansu, China. PS was obtained from the Lanzhou timber market in Gansu Province, China. All the samples were exposed to sunlight for 2 days to completely remove excessive moisture, and the samples were then shredded and sieved to <0.42 mm. The prepared samples were stored in a brown jar with a lid for subsequent analyses.

The proximate analysis, ultimate analysis, and lignocellulosic content analysis of the feedstocks were conducted as described in our previous study [29]. The thermophysical properties analysis of the samples was performed using a KD2 Pro thermal properties analyzer (Decagon Devices, Inc., Pullman, WA, USA) based on the transient hot wire method [30]. All experimental tests were completed in 2019 at Lanzhou Jiaotong University.

### 2.2. Pyrolysis Experiments and Thermogravimetric Analysis

The pyrolysis experiments of samples were conducted in an STA PT1600 analyzer (Linseis Messgeräte GmbH, Selb, Germany). A fine sample powder of 8~10 mg was placed in a small Al_2_O_3_ crucible and heated from room temperature (30 °C) to 800 °C at different heating rates of 5, 10, and 15 °C·min^−1^. Nitrogen was initially purged to create an inert atmospheric environment and to prevent undesirable oxidation of the samples. The weight loss and temperature changes during the pyrolysis process were monitored, and the thermogravimetric (TG) and derivative thermogravimetric (DTG) curves were obtained, which provided the calculation of conversion (α) efficiency of the sample based on the Equation (1):(1)$$\alpha =\frac{m_{0}-m_{T}}{m_{0}-m_{f}}$$
where *m*_0_, *m_T_*, and *m_f_* are the initial, instantaneous, and final masses of the samples, respectively. 

### 2.3. Kinetic Analysis 

For solid-state reactions, the non-isothermal pyrolysis kinetics of biomass can be represented using the following Equation (2) [31]:(2)$$\frac{d\alpha }{dt}=k\left(T\right)f\left(\alpha \right)=Aexp\left(\frac{E_{\alpha }}{RT}\right)f\left(\alpha \right)$$
where $\frac{d\alpha }{dt}$ is the conversion rate; *t* is the time (min); *k*(*T*) is the rate constant of decomposition; *T* is the temperature (K); *f*(*α*) is the reaction model; A is the pre-exponential factions (s^−1^); *Ea* is the activation energy (kJ/mol); and *R* is the universal gas constant (8.314 J mol^−1^ k^−1^). 

To obtain the kinetic parameters, various model-free methods are employed, including the FWO, Freidman, KAS, and Starink methods. These methods do not require the assumption of any reaction model or frequency factor, which is highly recommended by the Kinetics Committee of the International Confederation for Thermal Analysis and Calorimetry (ICTAC Kinetics Committee) [32]

The FWO method [32] was used to calculate the activation energy from the plot of ln *β* vs. 1/*T* for a given *α* following Equation (3):(3)$$\ln\left(\beta \right)=\ln\left[\frac{AE_{\alpha }}{RG_{\alpha }}\right]-5.331-1.052\left(\frac{E_{a}}{RT}\right)$$
where *β* is the heating rate in °C·min^−1^.

The Freidman method [33] is a type of linear differential method that was used to calculate the activation energy with the plot of ln $\frac{d_{\alpha }}{d_{t}}$ against 1/*T_α_* for a given *α* following Equation (4):(4)$$\ln\left[\frac{d_{\alpha }}{d_{t}}\right]=\ln\left[f\left(a\right)A_{\alpha }\right]-\frac{E_{\alpha }}{RT_{\alpha }}$$

The KAS method [34] was used to calculate the activation energy from the plot of ln [$\frac{\beta }{T^{2}}$] vs. 1/*T* for a given of *α* following Equation (5):(5)$$\ln\left[\frac{\beta }{T^{2}}\right]=\ln\left[\frac{AR}{E_{\alpha }\;g\left(\alpha \right)}\right]-\frac{E_{\alpha }}{RT}$$
where *g*(*α*) is the integral form of the conversion dependence function *f*(*α*).

The Starink [35] method was used to calculate the activation energy from the plot ln [$\frac{\beta }{T_{\alpha }^{1.92}}$] vs. 1/*T* during each conversion following Equation (6):(6)$$\ln\left[\frac{\beta }{T_{\alpha }^{1.92}}\right]=Constant-1.008\left(\frac{E_{\alpha }}{RT_{\alpha }}\right)$$

The pre-exponential factor (*A*) in the Arrhenius equation was calculated by Equation (7):(7)$$A=\beta \;E_{\alpha }\;exp\left(\frac{E_{\alpha }}{R\;T_{m}}\right)/R\;T_{m}^{2}$$
where *T_m_* is the maximum peak temperature of the DTG curves, °C.

### 2.4. Mechanism Model f(α) Prediction

With the reaction model, *f*(*α*) was predicted using the *z*-master plot method. Master plots are the theoretical curves that depend on the kinetic model of the reaction but are independent of the kinetic parameters *Eα* and *A*. In this method, if the theoretical master plots overlapped with the experimental master plots, and the reaction mechanism *f*(*α*) was selected as the most appropriate kinetic model (Table 1). 

*z*-Master plots are a combination of the integral and differential forms of the reaction model following Equation (8):(8)$$z\left(a\right)=\frac{f\left(a\right)\times g\left(a\right)}{f\left(0.5\right)\times g\left(0.5\right)}$$

The theoretical data were obtained by various solid-state reaction mechanisms as Equation (9), shown in Table 1:(9)$$z\left(a\right)={\left(\frac{T_{a}}{T_{0.5}}\right)}^{2}\frac{{\left(\frac{da}{dt}\right)}_{a}}{{\left(\frac{da}{dt}\right)}_{0.5}}$$
where $T_{0.5}$ and ${\left(\frac{da}{dt}\right)}_{0.5}$ are the temperature and conversion rates at the degree of conversion equivalent to 0.5, respectively, i.e., the means at 50% conversion.

### 2.5. Data Analysis 

The statistical significance of intrinsic physicochemical properties was determined by One-way analysis of variation (ANOVA) using SPSS 20.0 statistical analysis software (IBM, New York, NY, USA). The Pearson correlation analysis was used to explore the effects of physicochemical properties on pyrolysis characteristics and behavior of agroforestry waste.

## 3. Results and Discussion

### 3.1. Intrinsic Physicochemical Properties of Samples

Identifying the intrinsic physicochemical properties of RS, ATB, and PS involves proximate analysis, ultimate analysis, lignocellulosic contents analysis, and thermophysical parameters analysis, as shown in Table 2 and Table 3. It is clearly observed that different kinds of biomass, although consisting of the same major constituents, have different compositions, which may have distinct effects on the pyrolysis characteristics and mechanisms. From the proximate analysis, RS contained relatively higher contents of water and ash than those of the ATB and PS, but its fixed carbon content was relatively lower, as shown in Table 2. The moisture content of all the samples is less than 10% and considered suitable for pyrolysis [36]. This primarily because a lower moisture content in biomass usually increases the heating value of feedstocks and reduces the energy and time requirements to reach the pyrolysis temperature. Low ash content (<4%) is also a desirable parameter for biomass pyrolysis, since high ash content is associated with the limit of mass and heat transfer during the pyrolysis process [37]. In addition, some inorganic compounds (Ca and K) in ash are effective catalysts and can improve the efficiency of pyrolysis [24]. The volatile contents of RS and ATB biomass were not significantly different but were lower than that of the PS. The volatile matter of samples often contributes to the yield of bio-oil. Fixed carbon content increased in the order of RS (11.47 ± 0.23%) < PS (13.49 ± 0.16%) < ATB (15.04 ± 0.17%), which is highly correlated with the heating value and combustion time of biomass [11].

According to Table 2, it can be seen from the ultimate analysis that the contents of C and H in the woody-based biomasses (ATB and PS) were relatively higher than those in the herbaceous-based biomass (RS), while the contents of O and N were exactly the opposite. Higher percentages of C and H and a lower content of O in the biomass contributed to the improved performance of pyrolysis products (bio-oil) [38]. The content of N in RS, ATB, and PS does not exceed 1%. The low proportion of N helps to reduce the emission of nitrogen oxides during the pyrolysis process.

The lignocellulose analysis and thermophysical properties are shown in Table 3. The content of hemicellulose, cellulose, and lignin of the samples ranged from 11.08 ± 2.65 to 27.95 ± 5.13%, from 41.08 ± 2.26 to 54.69 ± 2.01%, and from 16.05 ± 3.08 to 30.73 ± 0.97%, respectively. Cellulose is the primary constituent of the agroforestry waste studied here. In addition, RS, with its high content of hemicellulose, usually produces higher yields of bio-oil or liquid. ATB and PS, with their high contents of lignin, usually result in a high yield of biochar. 

The thermophysical properties of RS, ATB, and PS are shown in Table 3. The values of thermal conductivity (*K*) of the three types of agroforestry waste vary over a small range, which is from 0.110 ± 0.0049 to 0.116 ± 0.0041 W·m^−1^·°C^−1^. There was no significant difference (*p* > 0.05) in *K*, indicating that the raw materials had little influence on the properties of thermal conductivity. RS has a lower heat capacity (*C*_0_) value and thermal diffusivity (*D*) value compared to ATB and PS. Differences in the *D* value of biomass can affect the heat transfer from the external surface to the internal parts of the particles, which, in turn, affects the kinetics of pyrolysis [39]. 

### 3.2. TG and DTG Curves

The TG and DTG curves of RS, ATB, and PS are presented in Figure 1. The TG and DTG profiles show that the decomposition of RS, ATB, and PS biomass occurred primarily in three distinct stages of mass loss. The initial mass loss stage (Stage I) is the stage of dehydration, which primarily occurs in the range of 30 °C to approximately 190 °C. In this stage, the weight loss curve slightly decreased owing to the release of bound and unbound moisture and the removal of some light-weight hydrocarbons [40]. 

The second stage (Stage II) is the devolatilization stage, which appeared in the temperature range from 190 °C to approximately 380 °C owing to the degradation of aliphatic chains and the release of volatiles from the decomposition of hemicellulose and cellulose [41,42]. In this stage, a rapid decrease in biomass weight was observed in the TG curves. The weight loss of all the samples exceeded 50%, indicating that the devolatilization stage is the primary stage of biomass pyrolysis. As shown from the DTG curves, tiny “shoulders” were observed for RS, ATB, and PS in the range of 250 to 263 °C, 260 to 303 °C, and 302 to 313 °C, respectively. This could be related to the decomposition of the hemicellulose component, since the decomposition of hemicellulose primarily occurred at 220–315 °C [43]. At temperatures between 295 and 360 °C, sharp peaks with a high rate of weight loss were observed in the DTG curves, owing to the decomposition of cellulose in the samples. 

The last stage (Stage III) is the char formation stage, which occurred when the pyrolysis temperature was higher than 380 °C. The formation of char was attributed to the decomposition of lignin, which are the most complex aromatic polymers and the most difficult components of biomass to be pyrolyzed [44]. Studies have shown that lignin is decomposed during the broad temperature region from 190–900 °C with a slower rate of decomposition compared to other components, resulting in no obvious peaks appearing in the DTG curves [45].

### 3.3. Pyrolysis Characteristics

The detailed and typical pyrolysis characteristics of biomass pyrolysis for different stages are shown in Table 4. The final temperature (FT_1_) and corresponding temperature range (R_1_) of the initial stage increased with the heating rate in the initial stage, while the weight loss rate (WL_1_) showed the opposite trend. Higher values of FT_1_ and R_1_ mean that higher temperature and more energy is required for the first stage of pyrolysis, while higher values of WL_1_ mean that more pyrolysis products are emitted in this stage. At a lower heating rate (5 °C·min^−1^), the heating of biomass particles increases gradually, resulting in more efficient heat transfer within and among the particles. This process led to a more complete release of moisture and light volatiles from the samples, which is a consequence of the value of WL_1_ being twice the value of the moisture content of the biomass. In contrast, when the rate of heating was 15 °C·min^−1^, there was a relatively small difference between the value of WL_1_ and the moisture content of the biomass. Additionally, the pyrolysis characteristics of different feedstocks show a remarkable difference. The FT_1_, R_1_, and WL_1_ values of RS are larger than those of ATB and PS. This implies that herbaceous materials (RS) require higher temperatures and more energy during the first stage of pyrolysis compared to woody materials.

In the second stage, the final temperature (FT_2_) and corresponding temperature range (R_2_) and the weight loss of the second process (WL_2_) also increased with the increasing rate of heating. The values of WL_2_ all exceeded 50%, confirming that the second stage is the main stage of biomass pyrolysis. The results could be a result of the simultaneous decomposition of the main components (including hemicellulose, cellulose, and lignin) by the high rate of heating, leading to a shift in the pyrolysis to a higher temperature region [46]. This is also confirmed by the higher peak devolatilization rate (P_rate_) at a higher rate of heating. For example, at 15 °C·min^−1^, the P_rate_ value of RS, ATB, and PS were 2.60, 1.76, and 2.23%·°C^−1^, respectively. These values were approximately three-fold higher than those at 5 °C·min^−1^, which were 0.90, 0.68, and 0.79%·°C^−1^, respectively. When the heating rate increases, the P_rate_ of the feedstock also tends to increase as higher thermal energy is provided to the sample, promoting better heat transfer between the surroundings and the interior of the sample, which in turn leads to a rapid increase in the decomposition rate of the feedstock. Unlike the initial stage, RS had lower values of FT_2_, R_2_, WL_2_, and P_T_ and higher value of P_rate_ value compared with those of ATB and PS. The difference can be interpreted by the fact that biomass has a heterogeneous structure and possesses a number of constituents requires different amounts of energy at different stages of pyrolysis.

In the final stage, the temperature range (R_3_) of sample pyrolysis varied between 398 and 447 °C and decreased as the rate of heating increased. Compared with R_1_ and R_2_, the temperature range of R_3_ was obviously wider. This is related to the relatively wide range of pyrolysis temperatures of the lignin component of the biomass. The weight loss (WL_3_) of the RS, ATB, and PS decrease as the heating rate increased, except for the RS at 10 °C·min^−1^. It was clearly observed that the value of WL_3_ was higher in the RS than in the ATB and PS, which was probably owing to the higher content of ash in the RS that catalyzed the pyrolysis of lignin [47]. The total weight loss (WL_total_) of the whole process decreases gradually with the increasing heating rate; therefore, a low heating rate facilitates maximum decomposition compared to a high heating rate. The residues of samples (RSS) of RS, ATB, and PS were determined to be in the range of 9.85~15.75%, 21.24~26.15%, and 18.66~22.55%, respectively. Variation in the sample RSS also indicated that there were differences in the potential for biomass pyrolysis.

### 3.4. Correlation between the Intrinsic Physicochemical Properties and Thermal Characteristics 

The correlation matrix in Figure 2 shows the relationships between the intrinsic physicochemical properties and the characteristic parameters of pyrolysis. The water content of the biomass positively correlated (*p* < 0.05) with the WL_1_ and WL_3_, except for the WL_1_ at 10 °C/min. This corroborates that the loss of weight in the first stage (WL_1_) is primarily caused by the loss of moisture in the biomass. The effect of water content on WL_3_ could be a result of the presence of water vapor at elevated pressures lowering the temperature needed for the decomposition of lignin [48], which resulted in more lignin decomposition and increased the weight loss in the third stage (WL_3_). The ash content was significantly negatively correlated (*p* < 0.05) with R_2_ at the highest heating rate (15 °C·min^−1^), but this correlation was not significant under the low (5 °C·min^−1^) and medium (10 °C·min^−1^) heating rates. Ash could play a catalytic role in the decomposition of biomass in this study, which improved the pyrolysis efficiency and narrowed the temperature range of the second stage. No clear relationship was found between the content of volatile matter and characteristics of pyrolysis, while a high content of volatile matter presented more bio-oil and syngas products. The fixed C content significantly negatively correlated with the peak devolatilization rate (P_rate_), primarily owing to the difficulty of degrading the fixed C and reducing the rate of pyrolysis [49]. Thus, the proximate composition can be regarded as being an indicator of the pyrolysis rate.

The contents of C and H significantly positively correlated with WL_2_, while the content of O significantly negatively correlated with WL_2_. This could indicate that the substances released in the second stage of pyrolysis were compounds rich in C and H. The components of O in the sample existed as a recalcitrant fraction, which had difficulty decomposing during the second stage [50]. The content of N significantly negatively correlated with P_T_, probably due to most of the N in agroforestry waste being bound in proteins, which decomposed at a lower temperature. In addition, the content of N significantly negatively correlated with the R_2_ and FT_2_ under the 15 °C·min^−1^ conditions. This implied that the high content of N in the biomass can significantly shorten the range of pyrolysis temperature and decrease the final pyrolysis temperature under the high heating rate.

The content of cellulose significantly positively correlated with WL_2_ (*p* < 0.05). Cellulose is the main component of biomass, and the decomposition of cellulose occurred during the second stage. The higher the content of cellulose, the greater the mass loss in the second stage. The content of hemicellulose significantly negatively correlated with P_T_, FT_2_, and R_2_, except at a low heating rate of 5 °C·min^−1^. This could be a result of the poor thermal stability of hemicellulose, which results in lower initial pyrolysis temperatures and a narrower range of thermal decomposition temperatures [51]. The contents of cellulose and lignin significantly positively correlated with R_2_, indicating that cellulose and lignin conjointly determine the second stage of the pyrolysis process. The content of lignin significantly positively and negatively correlated (*p* < 0.05) with the RSS and WL_total_, respectively. Hashimoto et al. also found that lignin is enriched in carbon relative to the carbohydrate components that result in the formation of more carbonaceous pyrolysis residues [52]. These biomasses with high pyrolysis residues are often recommended for biochar production. Thus, the lignocellulosic contents of agroforestry waste can be regarded as an indicator for combustion, pyrolysis, or gasification applications. 

A correlation analysis between the thermophysical properties of biomass and the thermal decomposition characteristics, as shown in Figure 2. The values *C*_0_ and *D* significantly negatively and positively correlated (*p* < 0.05) with P_rate_ when the heating rate was 10 °C·min^−1^ and 15 °C·min^−1^, respectively. A high *C*_0_ value indicated that more energy was needed during pyrolysis for the same increase in temperature, which is not conducive to P_rate_ increase. A high *D* value contributed to the quick spread of heat from high-temperature surfaces to the low-temperature centers of the samples, which led to a higher P_rate_. Thermal diffusivity also had a significant effect on WL_total_ and RSS when the heating rate was 10 °C·min^−1^ and 15 °C·min^−1^, respectively. A higher *D* value of the sample facilitated heat transfer, which resulted in more sufficient pyrolysis of the biomass and reduced the RSS.

### 3.5. Kinetic Parameter Analysis 

The pyrolysis kinetic parameters of RS, ATB, and PS were investigated using isoconversional model-free methods, including the FWO, Freidman, KAS, and Starink methods, in the primary stage of pyrolysis (at a conversion range of 0.1 to 0.8, with a step interval of 0.05). The appropriate kinetic model was estimated through the master-plot method.

#### 3.5.1. *Ea* Analysis

The values of RS, ATB, and PS are calculated from the linear model equations ln *β* vs. 1/*T*, ln $\frac{d_{\alpha }}{d_{t}}$ vs. 1/*T_α_*, ln [$\frac{\beta }{T^{2}}$] vs. 1/*T*, and ln [$\frac{\beta }{T_{\alpha }^{1.92}}]$ vs. 1/*T* based on the Equations (3)–(6) for the FWO, Freidman, KAS, and Starink methods, respectively, and are shown in Figure 3a–c, respectively. The values of *Ea* varied with the conversion fraction depending on the raw material, owing to the heterogeneous nature of lignocellulose. In addition, the *Ea* value was related to the chosen iso-conversional model-free methods. At conversions of 0.1, the *Ea* value of RS obtained by the FWO, KAS, Starink, and Freidman methods varied between the range of 10.39 and 17.93 kJ·mol^−1^. These values were obviously lower than those of ATB and PS, which varied in the range of 37.26 to 87.06 kJ·mol^−1^ and 20.85 to 82.51 kJ·mol^−1^, respectively. It is implied that RS biomass undergoes pyrolysis more quickly than those of ATB and PS at low conversion fractions. A sharp increase in the *Ea* could be observed after α = 0.15 for all the methods and gradually reached the maximum *Ea* value at the conversion of 0.4–0.5, 0.55–0.65, and 0.65–0.75 for RS, ATB, and PS, respectively. The fluctuation in *Ea* was caused by the involvement of parallel, complex, and competing reactions in an inert atmosphere and was related to the percentage of components present in biomass [53]. The reason for the maximum *Ea* was the simultaneous decomposition of cellulose, hemicellulose, and lignin, which requires a large amount of energy. Thereafter, with the complete decomposition of unstable hemicellulose and cellulose, the pyrolysis of the remaining individual lignin reduced the value of the *Ea*.

Compared with the *Ea* value calculated by the FWO, KAS, and Starink methods, the value obtained by the Freidman method was higher in the upward phase and lower in the downward phase. This difference could be a result of the different principles of calculation for the FWO, KAS, and Starink methods, which adopt linear integral methods for analysis based on the heating rate, while the Freidman method utilizes a linear differential method for analysis based on the conversion rate. As shown in Figure 1, RS, ATB, and PS were more sensitive to the heating rates. As the heating rate increased from 5 to 15 °C·min^−1^, the peak devolatilization rate of RS, ATB, and PS increased by a factor of 3 from 0.68~0.90%·°C^−1^ to 1.76~2.60%·°C^−1^. However, the changes in conversional fractions were less significant. Similar results were reported by Chong et al., where model-free methods, such as FWO and KAS, accurately predicted the kinetic predictions of horse manure [50]. In addition, integral methods, such as FWO, KAS, and Starink, are more reliable than differential methods, such as the Freidman method, because their results are less likely to be affected by experimental error. 

The minimum (*Ea*[min]), mean (*Ea*[mean]), and maximum (*Ea*[max]) values of *Ea* are shown in Appendix A. The *Ea*[mean] values determined by the FWO, Friedman, KAS, and Starink methods during the conversion process of RS were 167.15, 195.58, 177.90, and 178.60 kJ·mol^−1^, respectively; those of the ATB were 195.37, 234.95 207.19, and 207.57 kJ·mol^−1^, respectively; and those of the PS were 200.58, 236.45, 191.27, and 201.53 kJ·mol^−1^, respectively. It is worth nothing that *Ea*[mean] values determined for the three types of biomasses were different, which could indicate that the thermal decomposition mechanism of the whole pyrolysis process was different. The *Ea*[min] and *Ea*[max] values were obtained from Figure 3, which represent the easiest and most difficult stages of biomass in the pyrolysis process, respectively. The *Ea*[min] essentially occurred at the initial stage of pyrolysis, while the occurrence of *Ea*[max] was dependent on the feedstocks and the calculation model.

Figure 4 shows the correlation between the intrinsic physicochemical properties of the biomass and *Ea*[min], *Ea*[mean], and *Ea*[max]. Therefore, the effect of the intrinsic physicochemical properties of the biomass on *Ea* were analyzed. The *Ea*[min] values significantly positively correlated with the fixed C and *C*_0_ (*p* < 0.05) and negatively correlated with *D* (*p* < 0.05). High fixed C and *C*_0_ do not facilitate the start of biomass pyrolysis, but high thermal conductivity values are favorable to the start of biomass pyrolysis. Aup-Ngoen et al. noted that the rhizome biomass of cassava (*Manihot esculenta*) with high fixed C withstood more decomposition compared to the other types of biomass residues [54]. Large *C*_0_ values caused delayed heating of the particles owing to the increased thermal inertia and, therefore, were characterized by a larger *Ea* value [55]. In addition, pyrolytic heating is a surface heating process in which the heat is transported via outward to inward conduction. The high thermal diffusivity of the feedstock ensures that more heat can flow through the sample particles at a higher rate, thus increasing the energy efficiency and reducing the activation energy of the pyrolysis [30]. The thermophysical properties of biomass were also correlated with Ea[mean] and Ea[max], but not significantly. This indicated that the influence of the thermophysical properties of biomass on the kinetics of pyrolysis is mainly reflected in the initial stage.

The *Ea*[mean] and *Ea*[max] obtained by the FWO, Freidman, and Starink methods significantly positively correlated with the sum content of cellulose and lignin, indicating that the contents of cellulose and lignin determines the energy required for the pyrolysis process of agroforestry waste. It also confirmed that the *Ea*[max] during the pyrolysis process was caused by the simultaneous decomposition of cellulose and lignin, which requires a large amount of energy. The maximum activation energy values significantly negatively correlated with the contents of moisture, ash, N, and hemicellulose. The reason is that hemicellulose and the N-containing substances in biomass have a weak bond strength and are easily decomposed [56]. The catalytic role played by ash in the pyrolysis process helped to reduce the activation energy of the pyrolysis. 

#### 3.5.2. Pre-Exponential Factors (*A*) Analysis

The values of pre-exponential factors (*A*) for RS, ATB, and PS calculated by Equation (7) within the conversion ranges of 0.1 and 0.8 are provided in Appendix A. The *A* values of RS, ATB, and PS calculated from the FWO, Freidman, KAS, and Starink methods ranged between 5.36 × 10^−3^ and 2.62 × 10^19^ s^−1^ for RS, between 2.86 and 6.66 × 10^24^ s^−1^ for ATB, and between 5.55 × 10^−2^ and 4.00 ×10^25^ s^−1^ for PS. The enormous range of *A* signified the complicated chemical reactions that occurred during the pyrolysis process owing to the complex structure and composition of the RS, ATB, and PS feedstocks. In the conversion rate between 0.1 and 0.15, the A values of the three agroforestry wastes were less than 10^9^ s^−1^, indicating that only surface reactions occurred. When the conversion rates of RS, ATB, and PS are greater than 0.2 and 0.3 and the value of *A* is greater than 10^10^, it implies that the cellulose is starting to decompose. This also explains why the *Ea* of RS is higher than that of ATB and PS in the initial stage (α ≤ 0.2). Typically, *A* > 10^14^ s^−1^ infers that a high collision of molecules is required, and therefore, a high *Ea* is needed for the pyrolysis of biomass [36]. When the conversional fraction ranged from 0.45 to 0.50, 0.60 to 0.65, and 0.65 to 0.70 for RS, ATB, and PS, respectively, the pre-exponential value ranged from 4.48 × 10^17^ to 4.00 × 10^25^ s^−1^. This indicated that higher molecular collision requires more energy owing to the simultaneous decomposition of hemicellulose, cellulose, and lignin during this process. 

#### 3.5.3. Estimation of the Reaction Mechanism [*f*(α)] 

The master-plot method was utilized to discriminate the most probable mechanism of the biomass degradation process by comparing the theoretical curves with experimental data. The values of theoretical curves were obtained from Equation (8) by utilizing the solid-state reaction models *f*(*α*) and *g*(*α*) described in Table 1. The experimental data were determined based on the thermogravimetric data using Equation (9). Figure 5 presents the comparison between the theoretical master plot and the experimental data of RS, ATB, and PS. At the initial stage of pyrolysis for RS (0.15 ≤ *α* ≤ 0.30), ATB (*α* ≤ 0.15), and PS (0.10 ≤ *α* ≤ 0.25), the experimental data overlap or are close to the D1 curve. Similar results were described by Poletto et al. [57]. This indicates that at lower conversional fractions, heat is transferred throughout the sample by the process of diffusion (D1) [58]. The results also confirm the influence of the thermophysical properties of biomass on the initial stage of pyrolysis. 

When the conversion rate was in the range of 0.40 and 0.50, 0.20 and 0.35, and 0.30 and 0.45 for RS, ATB, and PS, respectively, they effectively coincided with the A4 curve. This is owed to the rupture of some ordered cellulose chains that formed compounds of low molecular mass, which could act as centers for random nucleation and growth for the degradation reaction. Small amounts of inorganic salt ash in biomass play a role in inducing the heterogeneous nucleation of the volatiles. With a further increase in conversion, the experimental data of RS coincided with the F3 plot in the range of α = 0.55–0.60, which could be caused by the pyrolysis of lignin. In this type of mechanism, the degradation begins at a random point that acts as a growth center for the development of the degradation reaction. The experimental data of ATB and PS effectively fit the D3 or R3 models in the range of 0.40 and 0.55 and 0.50 and 0.55, respectively. This mechanism is primarily a result of the high loss of mass in the sample, and the quick reduction of sample size caused the sample volume to contract. At a conversional fraction > 0.6, the experimental data of ATB and PS were close to the D2 plots, while there was no overlap between the experimental data of RS and the theoretical plots. This level of conversion involves the formation of biochar, which is a highly porous material and allows gases and volatiles to diffuse through its pores during the pyrolysis process. Thus, the Valensi diffusion model (D2) can be confirmed as the major reaction mechanism for the decomposition of ATB and PS owing to the diffusion of gases and volatiles through the pores of the produced char. Consistent with our results, Gogoi et al. also observed nucleation followed by diffusion as the main mechanisms for the pyrolysis of agricultural residues [59].

## 4. Conclusions

In this study, the effect of intrinsic physicochemical properties of agroforestry waste on its pyrolysis characteristics was comprehensively investigated. The results of the study showed that:(1)There were significant differences in the intrinsic physicochemical properties of the three agroforestry wastes. The rich lignocellulose content; low moisture, ash, and N element content; and good thermophysical properties indicate that the agroforestry waste has good prospects for pyrolysis.(2)The intrinsic physicochemical properties of agroforestry waste have a significant influence on their pyrolysis characteristics. Moisture is positively correlated with the weight loss in the first stage. The second stage is the main stage of pyrolysis, and the pyrolysis characteristics are influenced by the combination of several intrinsic physicochemical properties. The content of fixed carbon was positively correlated with P_rate_. WL_2_ was positively correlated with C and cellulose content and negatively correlated with O content. R_2_ was positively correlated with the sum of cellulose and lignin content. RSS was positively correlated with lignin content. In addition, C_0_ and D also had an effect on P_rate_, but not significantly at the lower heating rate conditions. This indicates that the intrinsic physicochemical properties of the biomass on the pyrolysis characteristics are also influenced by pyrolysis conditions.(3)The *Ea*[min] values were significantly positively correlated with *C*_0_ and negatively correlated with *D*. The *Ea*[mean] and *Ea*[max] were significantly positively correlated with the sum content of cellulose and lignin, indicating that the contents of cellulose and lignin determine the energy required for the pyrolysis process of agroforestry waste.(4)Vast variations in *A* value and several reaction models (diffusion model, growth model, and geometrical contraction model) involved in the pyrolysis process indicate that the pyrolysis of agroforestry waste is a complex process due to the heterogeneity of its intrinsic physicochemical properties.

In summary, based on these findings, it can be concluded that agroforestry wastes have good pyrolysis potential, and the pyrolysis of agroforestry waste is a complex process due to the heterogeneity of its intrinsic physicochemical properties. The qualitative and quantitative effects of these intrinsic properties on the pyrolysis characteristics and behavior obtained from this study will help researchers and agroforestry industries directly estimate pyrolysis rates and control the efficiency of the pyrolysis process.

## Figures and Tables

**Figure 1 materials-16-00222-f001:**
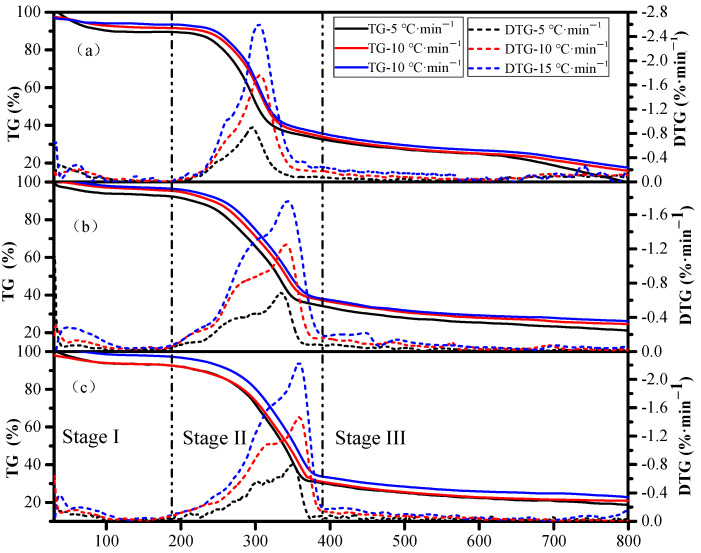
TG and DTG curves of for the pyrolysis of RS (**a**), ATB (**b**), and PS (**c**) at heating rates of 5, 10, and 15 °C·min^−1^.

**Figure 2 materials-16-00222-f002:**
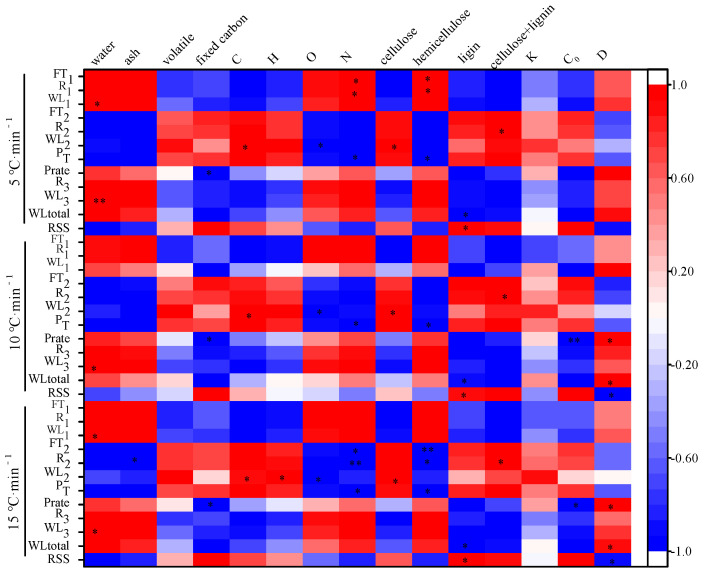
Correlation coefficient matrix of biomass composition and thermal characteristic parameters; ***** means *p* < 0.05, ****** means *p* < 0.01.

**Figure 3 materials-16-00222-f003:**
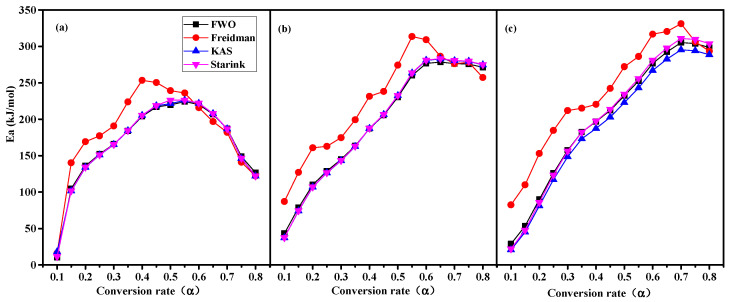
Variation in activation energy from different isoconversional model-free methods of RS (**a**), ATB (**b**), and PS (**c**).

**Figure 4 materials-16-00222-f004:**
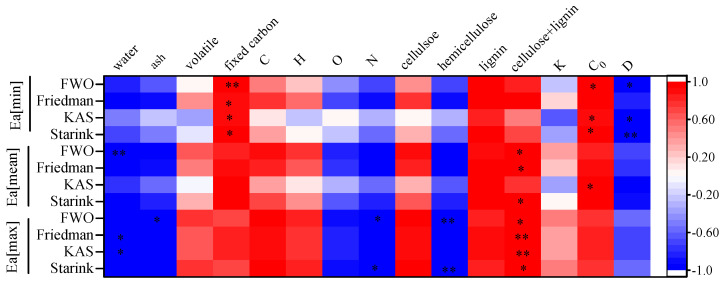
Correlation coefficient matrix of biomass intrinsic physicochemical properties and *Ea*[min], *Ea*[mean], and *Ea*[max]; * means *p* < 0.05, ** means *p* < 0.01.

**Figure 5 materials-16-00222-f005:**
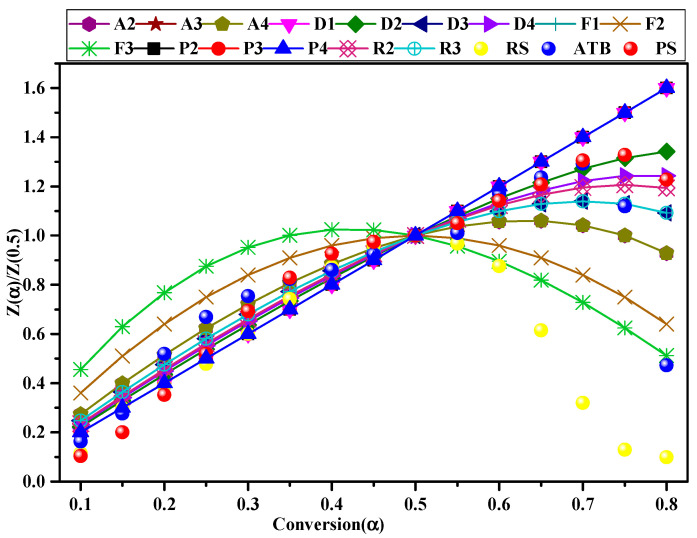
Master plot for theoretical models and experimental data variation with the conversion for RS, ATB, and PS.

**Table 1 materials-16-00222-t001:** Various theoretical kinetic reaction models [$f\left(\alpha \right)$] and their integral expression [*g*(α)] of solid pyrolysis process.

Kinetic Models		*f*(*α*)	*g*(*α*)
Nucleation models or growth model			
Power law	P2	$2\alpha^{1/2}$	$\alpha^{1/2}$
	P3	$3\alpha^{2/3}$	$\alpha^{1/3}$
	P4	$4\alpha^{3/4}$	$\alpha^{1/4}$
Avarami–Erofeev	A2	$2\left(1-\alpha \right){\left[-\ln\left(1-a\right)\right]}^{1/2}$	${\left[-\ln\left(1-\alpha \right)\right]}^{1/2}$
	A3	$3\left(1-\alpha \right){\left[-\ln\left(1-a\right)\right]}^{2/3}$	${\left[-\ln\left(1-\alpha \right)\right]}^{1/3}$
	A4	$4\left(1-\alpha \right){\left[-\ln\left(1-a\right)\right]}^{3/4}$	${\left[-\ln\left(1-\alpha \right)\right]}^{1/4}$
Diffusion model			
One-dimensional diffusion	D1	$\left(1/2\right)\alpha^{-1}$	$\alpha^{2}$
Two-dimensional (Valensi model)	D2	${\left[-\ln\left(1-\alpha \right)\right]}^{-1}$	$\left(1-\alpha \right)\ln\left(1-\alpha \right)+\alpha$
Diffusion control (Jander model)	D3	$(3/2){\left(1-\alpha \right)}^{2/3}/\left[2\left(1-{\left(1-\alpha \right)}^{1/3}\right)\right]$	${\left[1-{\left(1-\alpha \right)}^{1/3}\right]}^{2}$
Diffusion control (Ginstling model)	D4	$(3/2){\left[{\left(1-\alpha \right)}^{-1/3}-1\right]}^{-1}$	$1-\left(2/3\right)\alpha -{\left(1-a\right)}^{2/3}$
Order-based model			
First order	F1	1−α	$\alpha^{2}$
Second order	F2	${\left(1-\alpha \right)}^{2}$	$\left(1-\alpha \right)\ln\left(1-\alpha \right)$
Third order	F3	${\left(1-\alpha \right)}^{3}$	$1/2[{\left(1-\alpha \right)}^{-2}-1]$
Geometrical contraction model			
Contracting area	R2	$2{\left(1-\alpha \right)}^{1/2}$	$1-{\left(1-\alpha \right)}^{1/2}$
Contracting sphere	R3	$3{\left(1-\alpha \right)}^{2/3}$	$1-{\left(1-a\right)}^{1/3}$

**Table 2 materials-16-00222-t002:** Proximate analysis and ultimate analysis of samples.

Samples		RS	ATB	PS
Proximate analysis (%)	water	5.01 ± 0.08 ^a^	2.63 ± 0.12 ^b^	2.24 ± 0.33 ^b^
	ash	3.49 ± 0.03 ^a^	2.32 ± 0.02 ^b^	1.55 ± 0.02 ^c^
	volatile matter	80.02 ± 0.54 ^b^	80.0 ± 0.42 ^b^	82.72 ± 0.39 ^a^
	fixed carbon	11.47 ± 0.23 ^c^	15.04 ± 0.17 ^a^	13.49 ± 0.16 ^b^
Ultimate analysis (%)	C	42.24 ± 0.02 ^c^	46.19 ± 0.08 ^b^	51.27 ± 0.07 ^a^
	H	5.52 ± 0.03 ^c^	5.69 ± 0.06 ^b^	6.55 ± 0.01 ^a^
	O	47.79 ± 0.05 ^a^	45.25 ± 0.17 ^b^	40.33 ± 0.06 ^c^
	N	0.97 ± 0.01 ^a^	0.55 ± 0.01 ^b^	0.30 ± 0.02 ^c^

Different letters in the same line indicate significant differences (*p* < 0.05) among RS, ATB, and PS.

**Table 3 materials-16-00222-t003:** Lignocellulosic contents and thermophysical properties of samples.

Samples		RS	ATB	PS
Lignocellulosic contents (%)	cellulose	41.08 ± 2.26 ^c^	46.22 ± 1.30 ^b^	54.69 ± 2.01 ^a^
	hemicellulose	27.95 ± 5.13 ^a^	16.68 ± 0.55 ^b^	11.08 ± 2.65 ^c^
	lignin	16.05 ± 3.08 ^c^	30.73 ± 0.97 ^a^	26.43 ± 1.22 ^b^
Thermophysical properties	*K* (W·m^−1^·°C^−1^)	0.112 ± 0.005 ^a^	0.110 ± 0.005 ^a^	0.116 ± 0.004 ^a^
	*C*_0_ (MJ·m^−3^·°C^−1^)	0.539 ± 0.013 ^b^	0.652 ± 0.027 ^a^	0.606 ± 0.017 ^a^
	*D* (m^2^·s^−1^)	0.208 ± 0.006 ^a^	0.168 ± 0.005 ^c^	0.191 ± 0.002 ^b^

Different letters in the same line indicate significant differences (*p* < 0.05) among RS, ATB, and PS.

**Table 4 materials-16-00222-t004:** Characteristic temperatures associated with the mass losses in different stages of decomposition during the pyrolysis of RS, ATB, and PS at a different heating rates (°C·min^−1^).

Pyrolysis Stages		RS			ATB			PS		
		5	10	15	5	10	15	5	10	15
The initial stage	FT_1_ (°C)	176	178	190	165	170	180	160	162	172
	R_1_ (°C)	146	148	160	135	140	150	130	132	142
	WL_1_ (%)	10.39	8.34	6.46	7.16	4.35	3.4	6.79	6.69	2.34
The second stage	FT_2_ (°C)	353	362	380	373	380	395	378	380	402
	R_2_ (°C)	177	184	190	208	210	215	218	218	230
	WL_2_ (%)	54.06	55.28	58.3	57.3	57.62	58.88	62.2	63.37	64.69
	P_T_ (°C)	295.99	306.45	304.91	335.84	342.43	343.79	351.28	358.62	358.12
	P_rate_ (%/°C)	0.9	1.77	2.6	0.68	1.25	1.76	0.79	1.46	2.23
The final stage	R_3_ (°C)	447	438	420	427	420	405	422	420	398
	WL_3_ (%)	25.27	20.62	23.94	14.28	13.34	11.57	12.35	11.01	10.42
	WL_total_ (%)	90.15	84.25	88.71	78.75	75.31	73.85	81.34	81.07	77.45
	RSS (%)	9.85	15.75	11.29	21.24	24.69	26.15	18.66	18.93	22.55

FT_1_—the final temperature of the initial stage; R_1_—the temperature range of the initial stage; WL_1_—the weight loss of the initial stage; FT_2_—the final temperature of the second process; R_2_—the temperature range of the second process; WL_2_—the weight loss of the second process; P_T_—peak devolatilization temperature; P_rate_—peak devolatilization rate; R_3_—the temperature range of the final process; WL_3_—the weight loss of the final process; WL_total_—the total weight loss of the whole process; RSS—the residues of samples.

## Data Availability

Not applicable.

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
