# Peer review of "Influence of Intrinsic Physicochemical Properties of Agroforestry Waste on Its Pyrolysis Characteristics and Behavior"

_materials, 2022, doi:10.3390/ma16010222_

Round 1

Reviewer 1 Report

This manuscript presents a study where the pyrolysis characteristics and kinetic of rape, straw, apple,  tree branches,  pine  sawdust  were studied, which as representatives of agroforestry wastes. Pyrolysis were made under dynamic conditions (30 to 900 °C) at different heating rates 5, 10, and 15 °C·min. Correlation analysis showed that intrinsic physicochemical properties play distinct roles in different pyrolysis stage.

The manuscript is well written and structured and is interesting and it can be published in present form.

Author Response

Response to the Review report 1

MS No. materials-2033328

Title: Influence of intrinsic physicochemical properties of agroforestry waste on its pyrolysis characteristics and behavior

Comments and Suggestions for Authors

This manuscript presents a study where the pyrolysis characteristics and kinetic of rape, straw, apple,  tree branches, pine  sawdust  were studied, which as representatives of agroforestry wastes. Pyrolysis were made under dynamic conditions (30 to 900 °C) at different heating rates 5, 10, and 15 °C·min-1. Correlation analysis showed that intrinsic physicochemical properties play distinct roles in different pyrolysis stage.

The manuscript is well written and structured and is interesting and it can be published in present form.

Response: Thanks to the reviewer. We appreciate the reviewer for this kind recommendation.

Reviewer 2 Report

The manuscript has interesting results. Requires some minor adjustments.

Comments

Line 84, 87, correct (Brassica campestris L.)

Line 85 correct (Pinus sp.)

Materials and Methods, where were the tests performed? In which year?

Line 185, please provide full details of the manufacturer of the statistical software

Line 299, correct FT1 – the final etc.

Line 308, Figure 2, correct °C·min-1

Line 374, correct kJ·mol-1 etc.

References, please delete publications older than 10 years (10, 11, 13, 17-21, 27, 30, 35-39, 45, 48, 51, 52, 54, 55, 58, 60), especially for publications from the previous ages (13, 19-21, 36-38, 51, 60)

There is no publication year for publication no. 20

Author Response

Response to the Review report 2

MS No. materials-2033328

Title: Influence of intrinsic physicochemical properties of agroforestry waste on its pyrolysis characteristics and behavior

Comments and Suggestions for Authors

The manuscript has interesting results. Requires some minor adjustments.

Response: Thanks to the reviewer. We appreciate the reviewer for this kind recommendation.

1) Line 84, 87, correct (Brassica campestris L.)

Response: Thanks to the reviewer. Corrections have been made.

2) Line 85 correct (Pinus sp.)

Response: Thanks to the reviewer. Corrections have been done.

3) Materials and Methods, where were the tests performed? In which year?

Response: Thanks to the reviewer. Correction has been done, as “All experimental tests were completed in 2019 at Lanzhou Jiaotong University.” in revised manuscript.

4)Line 185, please provide full details of the manufacturer of the statistical software

Response: Thanks to the reviewer. We are very sorry for our negligence of the company name of the manufacturer of the statistical software. Correction has been made. The manufacturer of the statistical software “(IBM, New York, NY, USA)” has been added.

5) Line 299, correct FT1 – the final etc.

Response: Thanks to the reviewer. FT1 and FT2 represent the final temperatures of the first and second stages, respectively. We are sorry. No correction was made.

6) Line 308, Figure 2, correct °C·min-1

Response: Thanks to the reviewer. Corrections have been done.

7) Line 374, correct kJ·mol-1 etc.

Response: Thanks to the reviewer. Corrections have been done.

8)References, please delete publications older than 10 years (10, 11, 13, 17-21, 27, 30, 35-39, 45, 48, 51, 52, 54, 55, 58, 60), especially for publications from the previous ages (13, 19-21, 36-38, 51, 60)

Response: We appreciate the reviewer for this kind recommendation. References 11, 13, 18-23, 36-39, 51 and 60 have been deleted.

9) There is no publication year for publication no. 20

Response: Thanks to the reviewer. The publication year for publication no. 20 “1999” has been added.

Reviewer 3 Report

The processed paper is understandable. The topic under the investigation is relevant to be published. But there are several problems necessary to be solved before possible paper publishing.

1) The waste pyrolysis is already well discussed topic. The paper does not provide enough arguments to consider the topic as a really innovative.

2) The processed literature overview is limited. The knowledge gap is not well specified and explained.

3) There is no really strong hypothesis or research question defined. Also, the paper's objectives are not specified enough in detail.

4) The choice of the research methodology is not explained and discussed.

5) However, the results' part of the paper is interesting and well written. The discussion part of the paper is very limited. There is no relevant discussion related to already published papers or findings related to the topic under the investigation.

6) The final conclusion is to simple and brief. There are no relevant recommendations for future research activities. Also, paper's limitations are not specified. The paper originality and importance of final results are not highlighted. It is not clear for whom the paper should be considered as important one.

It is necessary to revise the paper before possible publishing.

Author Response

Response to the Review report 3

MS No. materials-2033328

Title: Influence of intrinsic physicochemical properties of agroforestry waste on its pyrolysis characteristics and behavior

Comments and Suggestions for Authors

The processed paper is understandable. The topic under the investigation is relevant to be published. But there are several problems necessary to be solved before possible paper publishing.

1) The waste pyrolysis is already well discussed topic. The paper does not provide enough arguments to consider the topic as a really innovative.

Response: Thanks to the reviewer. We have made correct according the Reviewer’s comments. The innovation of the article is mainly presented through two aspects. As described in introduction “The intrinsic physical and chemical properties of the feedstocks also have a vital influence on the pyrolysis behavior and kinetic parameters. Among the relevant characteristics of decomposition and the kinetics of biomass, the following intrinsic properties are critical: chemical composition (cellulose, hemicellulose, and lignin), proximate composition (water, volatile matter, ash, and fixed carbon), ultimate analysis (C, H, N, and O), and thermal properties (thermal conductivity, heat capacity, and thermal diffusivity) [19-23]. For example, the unique pyrolysis curve of biomass depends on its lignocellulose composition. The differences in proximate composition and thermophysical properties directly affect the activation energy, products and reactor throughput of biomass pyrolysis. In addition, the elemental composition of biomass can be used to predict reasonably well the thermodynamic properties of biomass, such as Gibbs free energy of formation. However, there are only a few attempts have been made to correlate the pyrolytic characteristics and kinetic of biomass with its intrinsic physical and chemical proper-ties, the information provided on the influence of physicochemical properties of raw materials on pyrolysis behavior is limited, especially regarding the pyrolytic behavior of agroforestry waste [24-26].” in the revised manuscript.

2) The processed literature overview is limited. The knowledge gap is not well specified and explained.

Response: Thanks to the reviewer. We have made correct according the Reviewer’s comments. As described in introduction “The influence of external factors has been extensively studied, such as heating rate, pressure, particle size and other factors on pyrolysis characteristics. The maximum rate of decomposition and the whole conversion rate of pyrolysis also increases with the in-creasing heating rates owing to the increase in thermal energy [16]. Increasing the pressure of the batch autoclave reactor not only increases the aromaticity and crystallinity of the pyrolysis products, but also leads to an increase in the Ea of the pyrolysis process [17]. The higher particle size ranges of biomass aslo resulted in relatively higher average Ea arising from the heat transfer limitation between particles [18]. Thus, these studies provide an im-portant theoretical scientific basis for tailoring the extra-particle factors to achieve high production efficiency and desired products.”.

3) There is no really strong hypothesis or research question defined. Also, the paper's objectives are not specified enough in detail.

Response: Thanks to the reviewer. Correction has been done. A really strong hypothesis or research question is posed, as described in introduction “The intrinsic physical and chemical properties of the feedstocks also have a vital influence on the pyrolysis behavior and kinetic parameters”.

The main objectives of this study to understand the influence of intrinsic properties of raw materials on the pyrolysis process and reaction mechanism for optimizing the design and amplification of the pyrolysis system. In addition, knowledge of the qualitative and quantitative effects of these intrinsic properties on the pyrolysis properties and behavior can help one to directly estimate the pyrolysis rate and control the pyrolysis efficiency of agroforestry wastes based on the intrinsic properties, instead of performing tedious empirical work. As described “Therefore, the primary objective of this study is to comprehensively analyze the intrinsic physicochemical properties of agroforestry waste, and to deeply understand the influence of intrinsic physicochemical properties of agroforestry on pyrolysis behavior and mechanism, which aims to provide a scientific basis for the efficient pyrolysis of agroforestry waste .” in the revised manuscript.

4) The choice of the research methodology is not explained and discussed.

Response: Thanks to the reviewer. The descriptions have been added, as described “ To obtain the kinetic parameters, various model-free methods are employed, including FWO, Freidman, KAS, and Starink methods. These methods dose not required assumption of any reaction model or frequency factor, which are highly recommended by the Kinetics Committee of the International Confederation for Thermal Analysis and Calorimetry (ICTAC Kinetics Committee)[30]’’ in the revised manuscript.

5) However, the results' part of the paper is interesting and well written. The discussion part of the paper is very limited. There is no relevant discussion related to already published papers or findings related to the topic under the investigation.

Response: Thanks to the reviewer. There are relatively few studies related to this topic. The relevant discussions of the published papers found have been cited to corroborate or explain the results of this experiment. For example, “ In first stage, the weight loss curve slightly decreased owing to the release of bound and unbound moisture, and the removal of some light-weight hydrocarbons [36]”, “The second stage (Stage II) is the devolatilization stage, which appeared in the temperature range from 190 °C to approximately 380 °C owing to the degradation of aliphatic chains and the release of volatiles from the decomposition of hemicellulose and cellulose [37,38]”, “The formation of char was attributed to the decomposition of lignin, which are the most complex aromatic polymers and the most difficult components to be pyrolyzed in bio-mass [4049]. Studies have shown that lignin is decomposed during the broad temperature region (190-900 °C) with a slower rate of decomposition compared, resulting in no obvious peaks appeared in the DTG curves [41]”, et al. We are sorry. No correction was done.

6) The final conclusion is to simple and brief. There are no relevant recommendations for future research activities. Also, paper's limitations are not specified. The paper originality and importance of final results are not highlighted. It is not clear for whom the paper should be considered as important one.

Response: Thanks to the reviewer for this kind suggestion. We have made correction according to the Reviewer’s comments, as “In summary, based on these finds, it can be concluded that agroforestry wastes have good pyrolysis potential, and the pyrolysis of agroforestry waste is a complex process due to the heterogeneity of its intrinsic physicochemical properties. The qualitative and quantitative effects of these intrinsic properties on the pyrolysis characteristics and behavior obtained from this study will help to the researchers and agroforestry industries directly estimate the pyrolysis rate and control the efficiency of the pyrolysis process.” was added in the last paragraph of the revised manuscript conclusion.

Reviewer 4 Report

12: "30 to 900 ° C)" remove the space before "°", Please make changes throughout the article.

30: In my opinion the abstract is too long, I suggest to remove some content, especially in the second part.

38: "year [1]." - In my opinion, Authors should write references in "[]", but without superscript.

229-230: "The thermogravimetric curve (TG) and derivative ones with the temperature curve (DTG)" - The authors have already explained these abbreviations.

Author Response

Response to the Review report 4

MS No. materials-2033328

Title: Influence of intrinsic physicochemical properties of agroforestry waste on its pyrolysis characteristics and behavior

Comments and Suggestions for Authors

1) Line 12: "30 to 900 ° C)" remove the space before "°", Please make changes throughout the article.

Response: Thanks to the reviewer. The mistakes in the whole article have been corrected.

2) Line 30: In my opinion the abstract is too long, I suggest to remove some content, especially in the second part.

Response: We appreciate the reviewer for this kind recommendation. A minor correction was been done.

3) Line 38: "year [1]." - In my opinion, Authors should write references in "[]", but without superscript.

Response: Thanks to reviewers for this kind suggestion. It is a uniform requirement for journals to use superscripts to write references. We are sorry. No correction was done.

4) Line 229-230: "The thermogravimetric curve (TG) and derivative ones with the temperature curve (DTG)" - The authors have already explained these abbreviations.

Response: Thanks to the reviewer. Corrections have been done.

Round 2

Reviewer 3 Report

The paper is acceptable for publishing. 

Author Response

Thanks to the reviewer.  We appreciate the reviewer for this kind recommendation.
